# Valid molecular dynamics simulations of human hemoglobin require a surprisingly large box size

Krystel El Hage[1], Florent Hédin[1], Prashant K Gupta[1], Markus Meuwly[1]*, Martin Karplus[2,3]*

[1]Department of Chemistry, University of Basel, Basel, Switzerland; [2]Department of Chemistry and Chemical Biology, Harvard University, Cambridge, Massachusetts, United States; [3]Laboratoire de Chimie Biophysique, ISIS, Université de Strasbourg, Strasbourg, France

**Abstract** Recent molecular dynamics (MD) simulations of human hemoglobin (Hb) give results in disagreement with experiment. Although it is known that the unliganded ($T_0$) and liganded ($R_4$) tetramers are stable in solution, the published MD simulations of $T_0$ undergo a rapid quaternary transition to an R-like structure. We show that $T_0$ is stable only when the periodic solvent box contains ten times more water molecules than the standard size for such simulations. The results suggest that such a large box is required for the hydrophobic effect, which stabilizes the $T_0$ tetramer, to be manifested. Even in the largest box, $T_0$ is not stable unless His146 is protonated, providing an atomistic validation of the Perutz model. The possibility that extra large boxes are required to obtain meaningful results will have to be considered in evaluating existing and future simulations of a wide range of systems.
DOI: https://doi.org/10.7554/eLife.35560.001

*For correspondence:
m.meuwly@unibas.ch (MM);
marci@tammy.harvard.edu (MK)

**Competing interests:** The authors declare that no competing interests exist.

## Introduction

Human hemoglobin is the paradigmatic model system for cooperativity in proteins. It transports oxygen from the lungs to the tissues and it is composed of two identical $\alpha$-chains ($\alpha 1$ and $\alpha 2$) and two identical $\beta-$chains ($\beta 1$ and $\beta 2$). They form two identical dimers ($\alpha 1\beta 1$ and $\alpha 2\beta 2$), whose relative orientation differs significantly in the unliganded ($T_0$) and liganded ($R_4$) tetramer. (*Baldwin and Chothia, 1979*) Although there is a vast literature on Hb and its cooperative mechanism (*Schechter, 2008*), how it functions at the atomistic level is still not fully understood (*Cui and Karplus, 2008*). The first insight into the mechanism was obtained from the low-resolution structures (5.5 Å) of the hemoglobin tetramer (*Muirhead and Perutz, 1963*), which showed that the heme groups were too distant to be able to interact directly. Monod, Wyman and Changeux formulated the allosteric (MWC) model (*Monod et al., 1965*; *Cui and Karplus, 2008*) based on the structural transition between two quaternary structures (T and R) to explain the indirect interaction between the heme groups required for cooperative oxygen binding.

Higher resolution (2.8 Å) X-ray structures of unliganded and liganded hemoglobin (*Perutz, 1970*) confirmed that there exist two quaternary structures (deoxy ($T_0$) and oxy ($R_4$)) for the tetramer and two tertiary structures for each individual chain (liganded and unliganded). Based on the structural results, as well as mutant data, *Perutz, 1970*) proposed a stereochemical mechanism for cooperativity, in which salt bridges (some with ionizable protons in the neutral pH range) provide the link between ligand-induced tertiary changes and the relative stability of the two quaternary structures. Shortly afterwards, the elements of the Perutz mechanism were incorporated into a structure-based statistical mechanical model (*Szabo and Karplus, 1972*). The model provides a quantitative

framework for the effects of specific tertiary structural changes induced by ligand binding on the relative stability of the T and R structures (*Szabo and Karplus, 1972*; *Gelin and Karplus, 1977*). The shift of the equilibrium from T to R as a function of ligand concentration results in the sigmoidal (cooperative) ligand binding curve.

Several papers using different force fields and simulation conditions have been published recently (*Hub et al., 2010*; *Yusuff et al., 2012*) describing molecular dynamics (MD) simulations of the T and R states, including $T_0$ and $R_4$ for which 1.25 Å resolution X-ray structures are available ($T_0$, 2DN2; $R_4$, 2DN3) (*Park et al., 2006*). Although $R_4$ was found to be stable for several hundred nanoseconds, the $T_0$ state was not. It was found to make a transition to an R-like state in the same time period (see also *Figure 1—figure supplement 2*). This occurs in spite of the fact that, experimentally, the $T_0$ state is about seven kcal/mol more stable than the $R_0$ state (*Edelstein, 1971*), which is derived from the ratio of the dissociation constants of liganded and unliganded Hb of $6.7 \times 10^5$ (*Thomas and Edelstein, 1972*).

Although the rate of the $T_0$ to $R_0$ transition has not been measured directly, it can be estimated from the experimentally determined $R_0$ to $T_0$ transition rates and the $R_0/T_0$ equilibrium constant. For the unliganded hemoglobin tetramer, the $R_0$ to $T_0$ rate is about 20 $\mu$s (*Sawicki and Gibson, 1976*). With the equilibrium constant of $[T_0/R_0]$ equal to $6.7 \times 10^5$ (*Edelstein, 1971*; *Thomas and Edelstein, 1972*), the $T_0$ to $R_0$ rate is estimated to be on the order of seconds with the major contribution to the activation barrier arising from the equilibrium free energy difference (7 kcal/mol) between $T_0$ and $R_0$. There is, thus, a striking disagreement between the transition time observed in the simulations and the estimate from experiment.

## Results and discussion

The instability of $T_0$ in the published simulations raises a fundamental question: What is wrong with them? In search for an answer, we focused on the hydrophobic effect (*Chothia et al., 1976*; *Lesk et al., 1985*), which arises from the disruption of the bulk water hydrogen bond network around nonpolar groups (*Rossky et al., 1979*; *Cheng and Rossky, 1998*). The theoretical analysis of Chandler and coworkers (*Chandler, 2005*; *Chandler and Varilly, 2012*) indicated that for large molecules, there was a 'dewetting' phenomenon that stabilizes a more compact structure. *Chothia et al. (1976)* noted that "A larger surface area is buried in deoxy- than in methemoglobin as a result of tertiary and quaternary structure changes. [..] This implies that hydrophobicity stabilizes the deoxy structure, the free energy spent in keeping the subunits in a low-affinity conformation being compensated by hydrophobic free energy due to the smaller surface area accessible to solvent.' Such a stabilizing effect of $T_0$ should appear naturally in a MD simulation, but evidently did not do so in the published simulations (*Hub et al., 2010*; *Yusuff et al., 2012*).

Having exhausted other possibilities (see Appendix for more details), we wondered whether the box size used for the MD simulations might be too small for the hydrophobic stabilization to be manifested. Because most of the simulation time is consumed by the waters, rather than the protein itself, which is of primary interest, a 'lore' has grown up about the minimal box size that can be used with periodic boundary conditions to carry out meaningful simulations. The standard requirement is that there must be at least five water molecules between any protein atom and the box boundary. (*Brooks et al., 1983*, *2009*) The box we first used was 75 Å; the $T_0$ tetramer dimensions are approximately $54 \times 49 \times 50$ Å, and there were 10,763 water molecules, as well as enough $Na^+$ and $Cl^-$ ions to yield a 0.15 m/L molar concentration. All MD simulations were done in the *NPT* ensemble (see SI for details).

To investigate the possibility that larger boxes were required for stabilizing $T_0$, we carried out $1\mu$s simulations with four cubic boxes, 75 Å (10,543 water molecules) 90 Å (20,756 water molecules), 120 Å (53,287) and 150 Å (105,073), see *Figure 1—figure supplement 1*. In all these simulations, His146$_{\beta1}$ and His146$_{\beta2}$, which play an essential role in the Perutz model (*Perutz, 1970*), were protonated. A comprehensive overview of the structural changes observed in the simulations of the four box sizes is provided in *Figure 1*. The essential result is that $T_0$ is stable for the entire simulation in the 150 Å box, while it is not in the smaller boxes (for details, see SI). *Figure 2* shows the structures obtained at the end of the 1 $\mu$s simulations, superposed on the X-ray structure that is more similar; that is, the oxy $R_4$ structure (2DN3) for the 90 and 120 Å simulations and the deoxy $T_0$ structure (2DN2) for the 150 Å box simulation (see also *Table 1*).

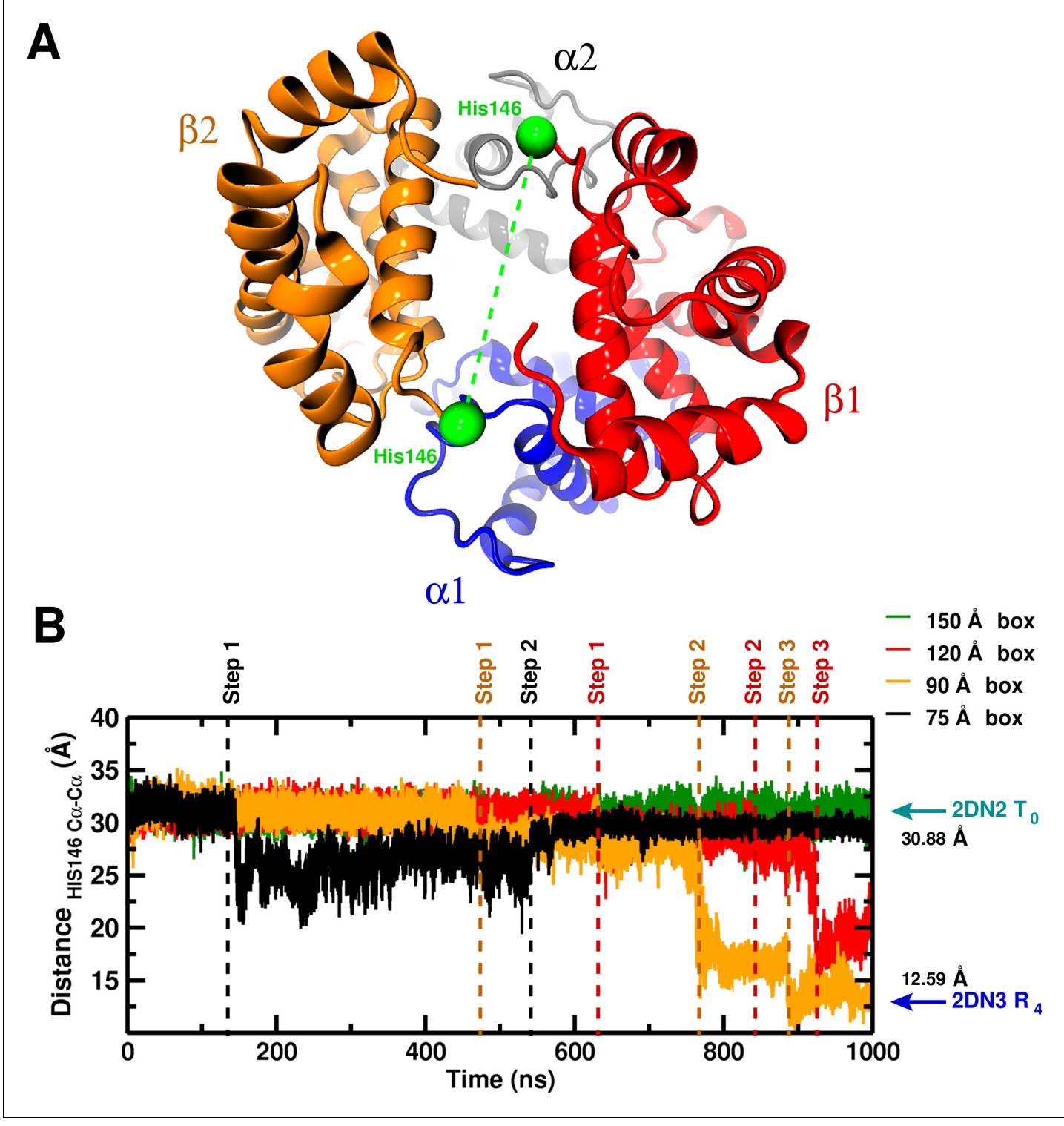

**Figure 1.** Global structural changes depending on box size. Panel A: Overall structure of the $\alpha1\beta1\alpha2\beta2$ hemoglobin tetramer. The His146 side chains (green spheres) are specifically indicated. Panel B: Temporal change of the $C\alpha$–$C\alpha$ distance between His146$_{\beta1}$ and His146$_{\beta2}$. The dashed lines (black, orange, red) indicate transition points for the 75, 90, and 120 box, respectively. Cyan and blue arrows indicate the values of the corresponding observable found for the deoxy $T_0$ (2DN2) and oxy $R_4$ (2DN3) states, respectively.

DOI: https://doi.org/10.7554/eLife.35560.002

The following figure supplements are available for figure 1:

**Figure supplement 1.** Tetrameric hemoglobin solvated in boxes of different sizes.

*Figure 1 continued on next page*

*Figure 1 continued*

DOI: https://doi.org/10.7554/eLife.35560.003

**Figure supplement 2.** Global structural changes depending on box size.

DOI: https://doi.org/10.7554/eLife.35560.004

**Figure supplement 3.** Local structural rearrangement of the C-terminus of the $\beta$ chain.

DOI: https://doi.org/10.7554/eLife.35560.005

**Figure supplement 4.** Local structural changes around His146.

DOI: https://doi.org/10.7554/eLife.35560.006

**Figure supplement 5.** Structural transition details.

DOI: https://doi.org/10.7554/eLife.35560.007

**Figure supplement 6.** Temporal structural changes of the $C\alpha$–$C\alpha$ distance between $His146_{\beta1}$ and $His146_{\beta2}$ in the 75 Å box for different simulation conditions.

DOI: https://doi.org/10.7554/eLife.35560.008

If the free energy difference between the $T_0$ and $R_0$ states (7 kcal/mol, see above) were to arise entirely from the relative stability of the water-water network, this value corresponds to an energy difference of $10^{-4}$ kcal/mol per water molecule when comparing the 90 and 150 Å boxes, which differ by $\approx 80,000$ water molecules. Such small energy differences are very difficult to capture in MD simulations and is not attempted here. However, it is interesting to note that the average number of hydrogen bonds per water molecule, $<N_{H-bonds}>$/molecule (see *Figure 3*) shows such an effect: for the three smaller boxes the $<N_{H-bonds}>$/molecule decreases by $10^{-3}$ to $10^{-4}$ with every transition. This is consistent with the estimated energy change per water molecule. Furthermore, *Figure 3* demonstrates that the fluctuation of $<N_{H-bonds}>$/molecule decreases with increasing box size which is the behaviour expected from statistical mechanics. It should be pointed out that the running averages were evaluated over time intervals between which transitions were observed, see *Figure 1*.

Based on hydrodynamic arguments, *Yeh and Hummer (2004)* showed that the water self-diffusion coefficient, $D$, calculated from an MD simulation of pure water (without ions) in a periodic box, scales as $N^{-1/3}$, where $N$ is the number of particles. For the largest box they studied (40 Å) the size correction was negligible. Interestingly, our simulation of the 75 Å box containing Hb yielded a value of $D$ that is much too small ($D = 4.25 \times 10^{-5}$ cm$^2$/s vs $D = 5.95 \times 10^{-5}$ cm$^2$/s); the latter is the correct value for the TIP3P water model. In *Figure 4A*, we show the results for the value of $D$ as a function of box size, plotted versus $1/L$ (nm$^{-1}$). As expected, all the pure water boxes are large enough so the calculated value agrees with the extrapolated value of Yeh and Hummer ($D = 5.9 \times 10^{-5}$cm$^2$/s) within statistical error. However, the calculated self-diffusion coefficient from MD simulations with Hb present, is identical to that of pure water only for the largest 150 Å box.

The results described here suggest that the correct free energy of hemoglobin, at least to the extent that $T_0$ is stable relative to $R_0$, requires that the simulation be done in a box large enough so that the water environment behaves like bulk water. In *Figure 4B*, we plot the calculated $D$–value for boxes containing Hb versus the time in ns when the first transition from the $T_0$ structure takes place. As can be seen, there is a linear relationship between the two. Of most interest is the fact that an extrapolation of the line indicates that in the 150 Å box, the first transition away from $T_0$ should take place at 700 ns. However, we have shown that in the 150 Å box, $T_0$ is still present at 1.4 $\mu$s. This provides strong evidence that in a 150 Å box, $T_0$ is in fact stable. This linear dependence was not expected. It is an interesting result whose origin, though, still requires explanations at a molecular level. The result that the lifetime of the $T_0$ state increases systematically with the increase in box size effectively corresponds to multiple simulations. The T-state was finally found to be stable in the 150 Å box for 1.4 $\mu$s, significantly longer than the extrapolated lifetime value (700 ns). The idea that $\mu$s-plus simulations are needed has become a 'lore' (similar to the box size-related 'rule' investigated here) with the availability of bigger computers, even when they are not required for a particular problem, as is the case here.

Simulations for all box sizes have also been done with His146 deprotonated. The results for that system showed that $T_0$ is stabilized in larger boxes, but after less than 100 ns a transition to an R-like state occurs. This provides direct evidence that His146 protonation is essential for stabilizing $T_0$, in accord with the Perutz model (*Perutz, 1970*).

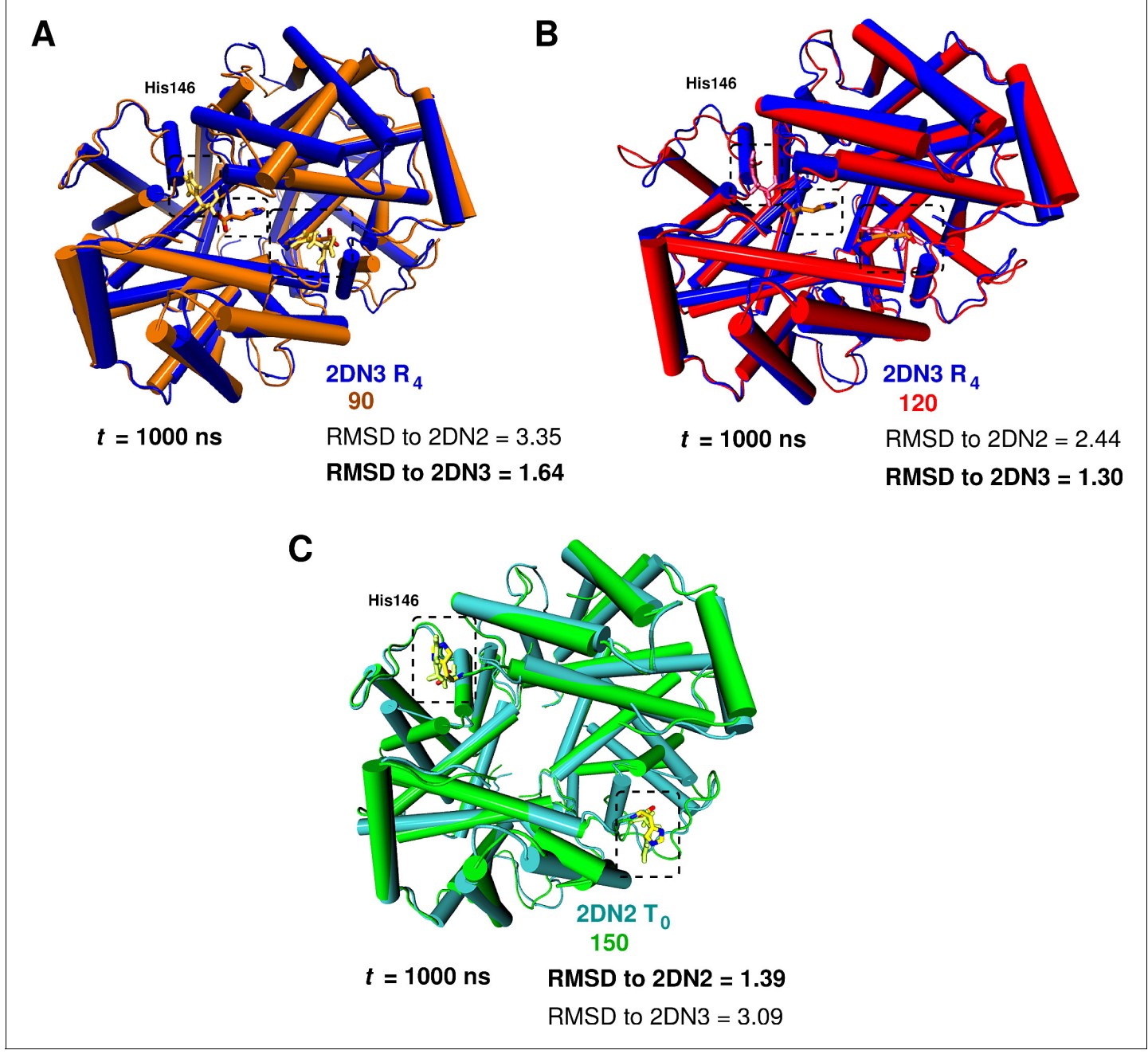

**Figure 2.** Global conformational rearrangement of tetrameric hemoglobin as a function of the box size. (**A**) Superposition of the 2DN3 structure and the Hb structure in the 90 Å box at $t = 1000$ ns. (**B**) Superposition of the 2DN3 structure and the Hb structure in the 120 Å box at $t = 1000$ ns. (**C**) Superposition of the 2DN2 structure and the structure in the 150 Å box at $t = 1000$ ns. In each case the RMSDs (in Å) to the 2DN2 and 2DN3 are also given.

DOI: https://doi.org/10.7554/eLife.35560.009

To relate the above results to the hydrophobic effect, we use the construct of Chandler (*Chandler, 2005*), who showed (*Figure 5A*) that significant water depletion around a spherical hydrophobic solute is expected when its radius is larger than 1 nm (10 Å). Since Hb is nearly spherical with a 2.8 nm radius, we calculated the $g(r)$ for the 90 and 150 Å box (see *Figure 5B*). The behavior in the 150 Å box is consistent with the expected water structuring, while for the 90 Å box it is not.

**Table 1.** $C\alpha$ RMSD (in Å) relative to the 2DN2 ($T_0$) and 2DN3 ($R_4$) X-ray structure (**Park et al., 2006**) of the end point structures at 1 $\mu$s.
In bold structure to which a computed Hb structure is closest.

| Structure | $C_\alpha$-$C_\alpha$ RMSD to 2DN2 | $C_\alpha$-$C_\alpha$ RMSD to 2DN3 |
|---|---|---|
| 2DN2 | – | 2.43 |
| 2DN3 | 2.43 | – |
| 75 Å box | 2.37 | 2.59 |
| 90 Å box | 3.35 | 0.**64** |
| 120 Å box | 2.44 | 0.**30** |
| 150 Å box | 0.**39** | 3.09 |

DOI: https://doi.org/10.7554/eLife.35560.010

The full radial distribution function is reported in **Figure 5—figure supplement 1** and **Figure 5—figure supplement 2B** shows the number of water molecules derived from it. It is found that up to a distance of 8.5 Å from the central cylinder ~150 water molecules are present which is consistent with explicit counting, see **Figure 5—figure supplement 2A**, and **Figure 5—figure supplement 3** for the corresponding probability distribution function. Furthermore, the structural transitions are accompanied by pronounced dewetting and water penetration as shown in **Figure 5—figure supplement 4** for the 90 Å box.

To summarize, the $T_0$ state is only found to be thermodynamically stable if (i) the hydration water behaves like bulk water as judged from the self-diffusion coefficient, (ii) the number of hydrogen bonds per water molecule is large enough and its fluctuation around the average sufficiently small, see **Figure 3—figure supplement 1**. Therefore, if water is not engaged in an adequate number of water-water H-bonds, solvent water is prone to attack the protein salt bridges, destabilizing them and eventually to break them.

As a more local measure of the effect of undersolvation the radial distribution function $g(r)$ around the (His146$_{\beta1}$)CG–O(water) was determined for all box sizes, see **Figure 6**. For the 150 Å box the $g(r)$ remains almost invariant whereas for the 120 Å box an appreciable change occurs during the time when the major structural transition at 920 ns takes place. For the smallest boxes, the local $g(r)$ is very variable, which supports the importance of locally structured water molecules for stabilizing the $T_0$ state.

In addition to the global behavior, local structural changes involving the two His146 residues were examined. The results (**Figure 1—figure supplement 3–5**) show that for the largest box, the parameters examined (e.g. the salt bridge to Lys40, discussed by Perutz) fluctuate around the equilibrium values near those of the $T_0$ structure while for the smaller boxes there are abrupt changes, correlated with global structural transitions. The transition away from the $T_0$ structure extends over approximately 10 ns during which several important contacts are broken or formed, see **Figure 1—figure supplement 5A**. This value suggests that the apparent activation energy from the simulations in the smaller solvent boxes is near zero, in contrast to an estimate on the order of seconds for the transition if the barrier arose from the 7 kcal/mol difference in stability.

Importantly, the ability of the 150 Å box to stabilize the T-state starting from the R-like structure in the 120 Å box after 1 $\mu$s was also explored. Solvating this structure in the 150 Å box, minimizing, heating and equilibrating it (see Materials and methods) and following the equilibrium dynamics for 1 $\mu$s, the final RMSD compared to the 2DN2 (T-state) structure is 1.97 Å (2.55 compared to 2DN3 (R-state)) starting from an RMSD of 2.73 Å (with respect to 2DN2) and 1.42 Å (2DN3) at the beginning of the simulation (after heating and equilibration), respectively. Concomitantly, the His146$_{\beta1}$–His146$_{\beta2}$ and His143$_{\beta1}$–His143$_{\beta2}$ distances change from values typical for an R-state structure (both ~12.6 Å) to 14.0 Å (for His143) and 18.0 (for His146) and approach separations indicative of a T-state structure (18.6 Å and 30.9 Å) without, however, fully completing the transition to a T-state structure. As mentioned above, the $R_0$ to $T_0$ transition time is about 20 $\mu$s, much longer than the simulation time.

Given the increasing use of molecular dynamics simulations to study conformational transitions in large proteins and in an explicit solvent environment, the present result that much larger boxes than

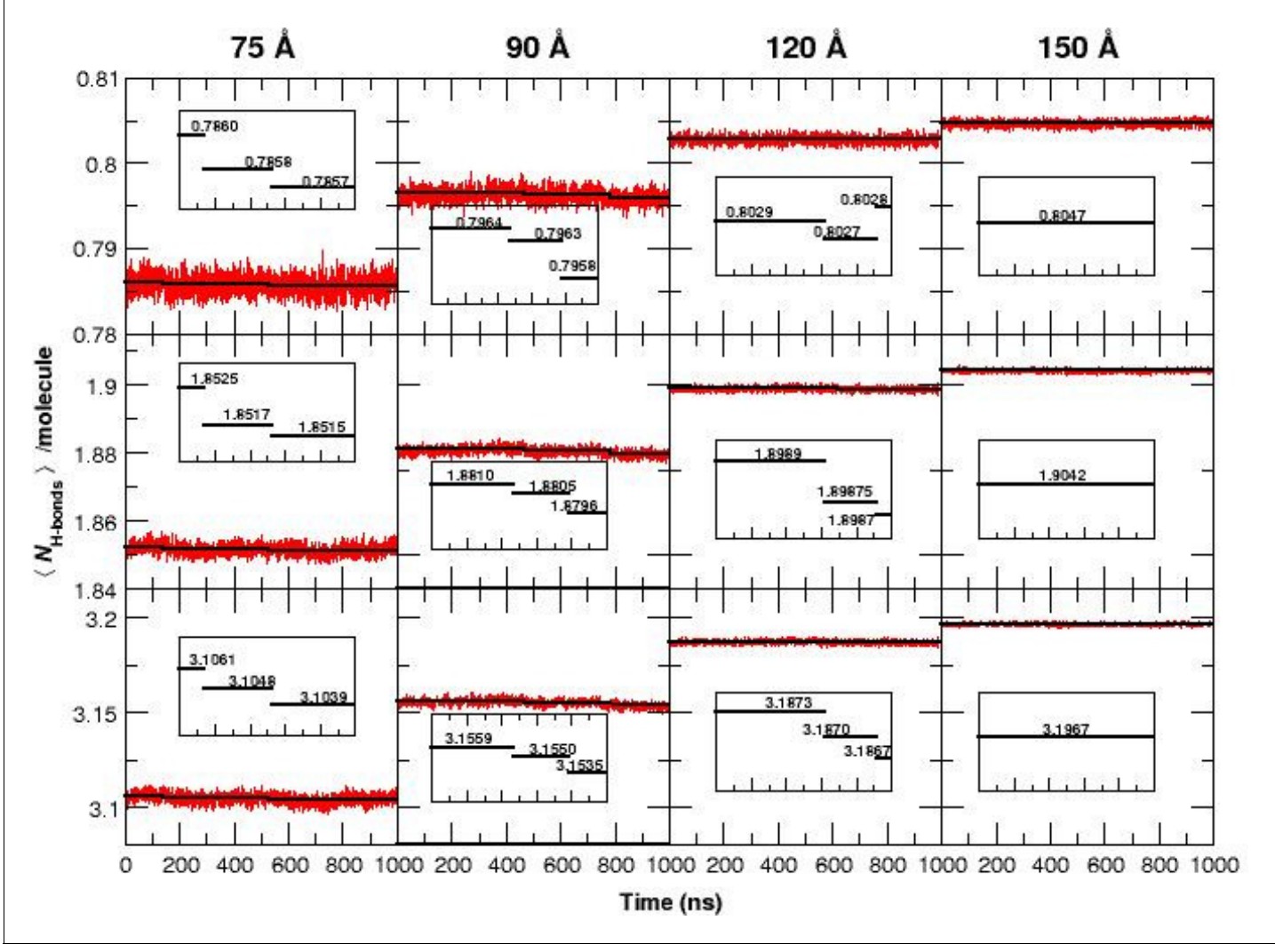

**Figure 3.** The average number of H-bonds per water molecule $<N_{H-bonds}>$/molecule, from analyzing water-water hydrogen bonds, during 1 $\mu$s MD simulations for all four box sizes and for three different donor-acceptor distance cutoffs: 2.8 (strong, top panels), 3.0 (medium, middle panels) and 3.3 Å (weak, bottom panels). The solid black lines are averages for time intervals corresponding to the lifetime of each state in each of the simulation boxes (see *Figure 1*): For the 75 Å box: 0 to 140 ns (first transition), 140 to 530 ns (second transition) and 530 to 1000 ns. For the 90 Å box: 0 to 470 ns, 470 to 780 ns and 780 to 1000 ns. For the 120 Å box: 0 to 620 ns, 620 to 920 ns, and 920 to 1000 ns. For the 150 Å box, the average is over the entire simulation since no transition occurs. The average reduction in $<N_{H-bonds}>$/molecule is maximum (0.1) when weak H-bonds are included (cutoff 3.3 Å, bottom row) and smallest (0.015) if only strong H-bonds (cutoff 2.8 Å, top row) are analyzed. In the 120 Å box (third column) and for the strong H-bonds the loss in $<N_{H-bonds}>$/molecule is insignificant but clearly increases when weak H-bonds are included. We also observe a significant decrease in the fluctuation of $<N_{H-bonds}>$/molecule between simulations in the smallest and the largest box sizes and a pronounced drop in $<N_{H-bonds}>$/molecule for the 75 and 90 Å boxes prior to the transitions, see insets.

DOI: https://doi.org/10.7554/eLife.35560.011

The following figure supplement is available for figure 3:

**Figure supplement 1.** Number $N$ of H-bonds and their fluctuations $\delta N$ for pure water and simulations including the protein for the four box sizes.

DOI: https://doi.org/10.7554/eLife.35560.012

those used standardly are required for what appears to be the hydrophobic effect to be manifested is of general interest. It has wide ranging implications for the interpretation and validity of previous simulations, as well as those to be undertaken in the future. Given that the magnitude of the effect is expected to depend on the size (and shape) of the molecule and its hydrophobicity, as well as possibly other properties (*Chandler, 2005*), the requirement for the use of larger boxes in simulations will have to be investigated in each case. One particularly relevant situation where capturing

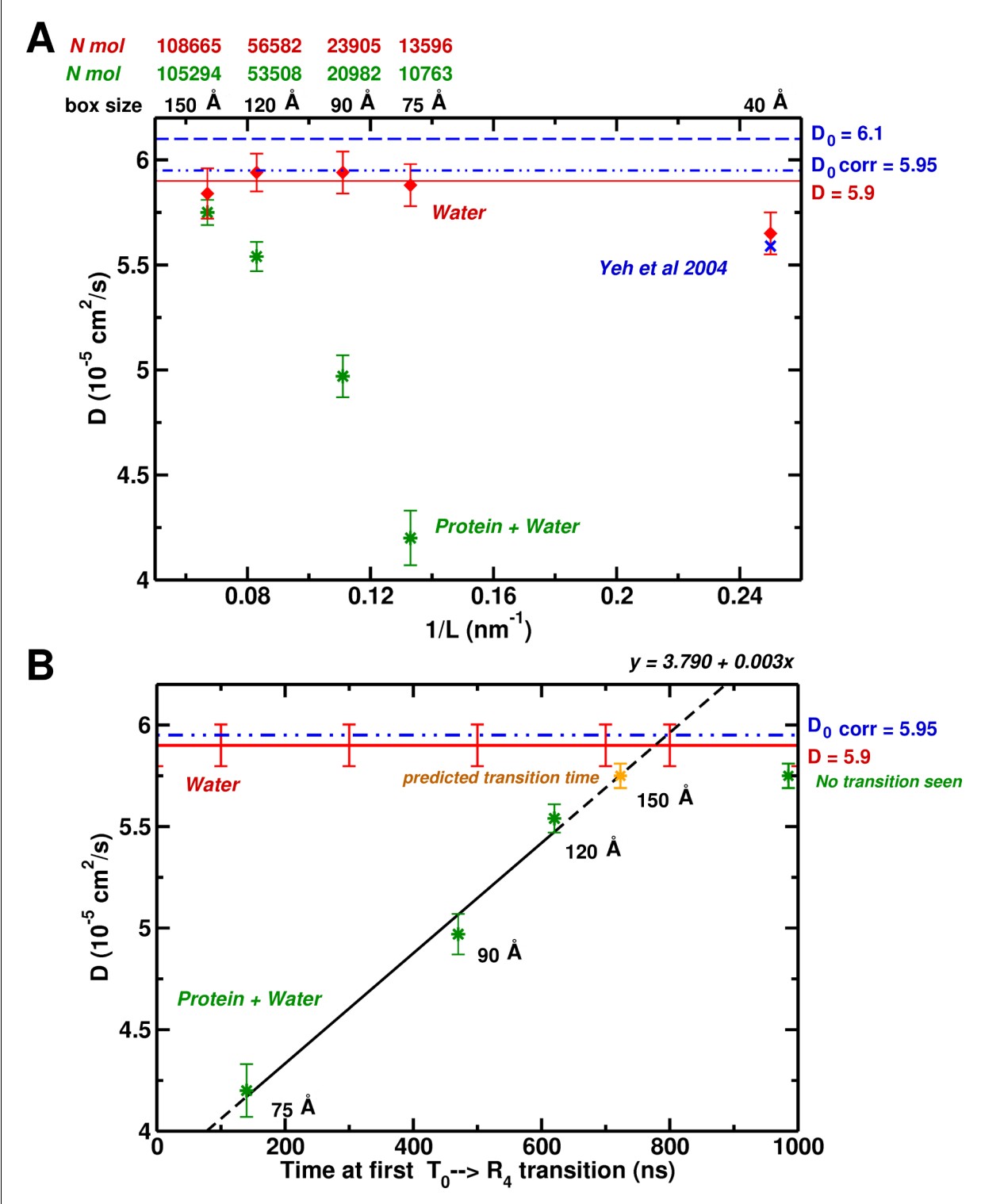

**Figure 4.** System-size dependence of the water diffusion coefficient. (**A**) Water diffusion coefficients *D* as a function of system size for systems with and without protein. Also, results from Hummer et al. are shown for comparison and validation. (**B**) The water *D* as a function of time for the first instability to occur. For the 75, 90, and 120 Å boxes instabilities were observed (see *Figure 1*) and scale linearly with the water *D*. The yellow symbol for the 150 Å box is the extrapolated value (700 ns).

DOI: https://doi.org/10.7554/eLife.35560.013

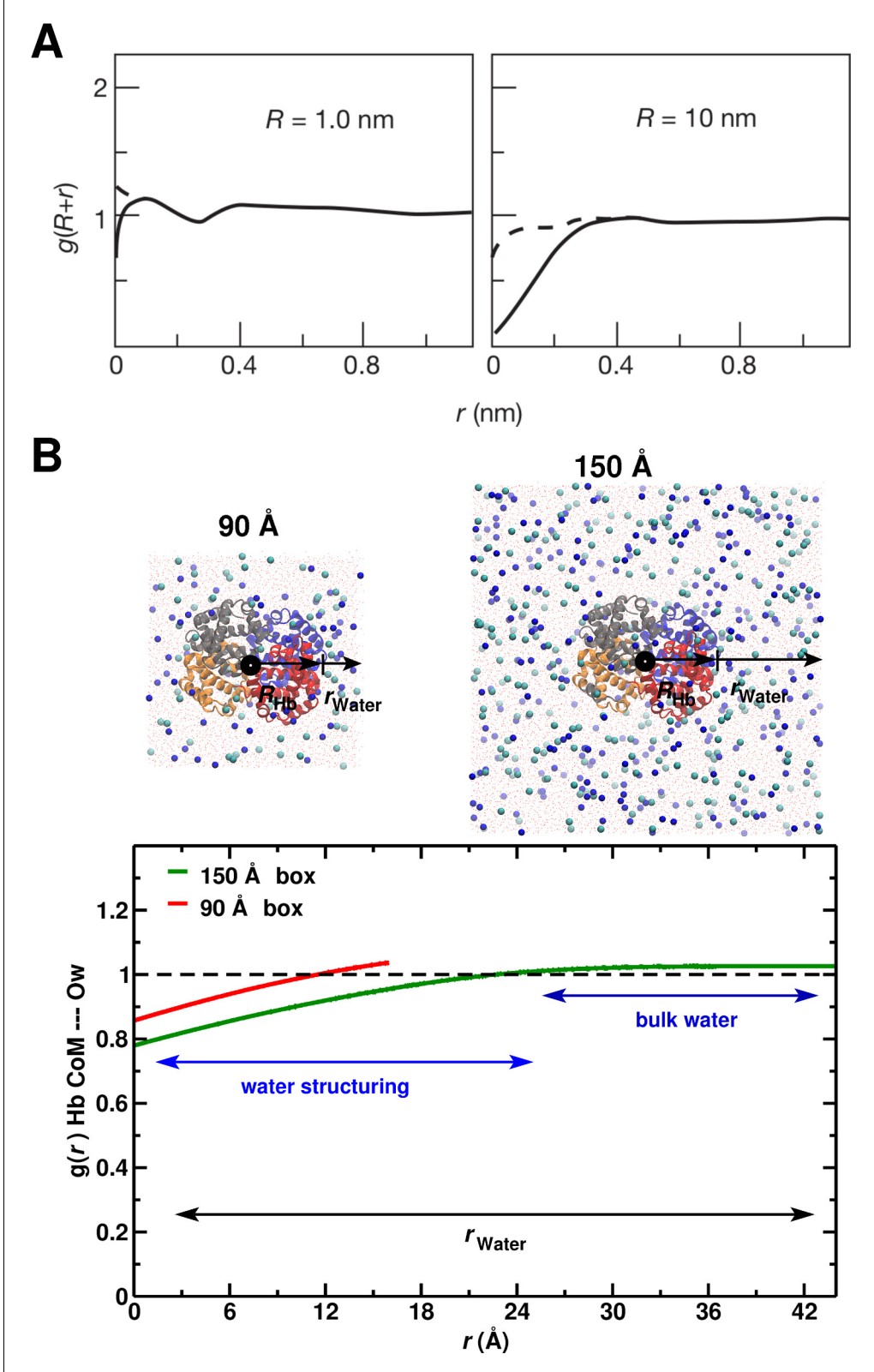

**Figure 5.** System-size dependence of the average equilibrium water density. (**A**) From Ref. (*Chandler, 2005*) (*Figure 3*); the average equilibrium water density a distance $r + R$ from spherical cavities. $R$ is the (spherical) size of the solute. Solid lines for the ideal hydrophobe, dashed line when van der Waals interactions are present. (**B**) Results for Hb in water for the 90 Å (red) and 150 Å (green) box. The radius of Hb (2.8 nm) is intermediate between the two cases in panel A.

*Figure 5 continued on next page*

*Figure 5 continued*

DOI: https://doi.org/10.7554/eLife.35560.014

The following figure supplements are available for figure 5:

**Figure supplement 1.** Enhanced version of *Figure 5* from the main MS.

DOI: https://doi.org/10.7554/eLife.35560.015

**Figure supplement 2.** Number of water molecules in the central cylinder of Hb.

DOI: https://doi.org/10.7554/eLife.35560.016

**Figure supplement 3.** Distribution $P(N)$ of water molecules in the central cylinder for the different box sizes.

DOI: https://doi.org/10.7554/eLife.35560.017

**Figure supplement 4.** Water at the $(\alpha 1\beta 2){:}(\alpha 2\beta 1)$ interface.

DOI: https://doi.org/10.7554/eLife.35560.018

the correct diffusional dynamics of the environment will play a crucial role is that in atomistic simulations for the crowded conditions (*Feig et al., 2017*) that exist in entire cells or parts thereof (*Zhou et al., 2008*). Possible errors in molecular dynamics simulation results for other phenomena, such as protein folding, for example, as well as for polymeric materials more generally have to be considered as well.

## Materials and methods

The influence of solvent layers on the structural stability of hemoglobin tetramer is investigated using Molecular Dynamics (MD) simulations. Extended large-scale simulation were performed with the CHARMM36 all atom force-field (*Best et al., 2012*) and the TIP3P water model (*Jorgensen et al., 1983*) using version 5.1.4 of the GROMACS package (*Abraham et al., 2015*) on GPUs.

The coordinates of the starting structure are taken from the X-ray structure of tetrameric human hemoglobin in the deoxy form, PDB code 2DN2 (1.25 Å resolution) (*Park et al., 2006*). The protonation states of the histidines were based on the 2013 study of Zheng et al. and the terminal $\beta$ histidines (His146) were both doubly protonated (*Zheng et al., 2013*). The protein was solvated in four different cubic boxes of increasing size: 75, 90, 120 and 150 Å. The system was neutralized by adding counter ions and the salt concentration of 0.15 m/L was achieved using $Na^+$ and $Cl^-$. The total number of atoms is: 39,432 for the 75 Å box, having 10,543 water molecules and 42 $Na^+$ and 38 $Cl^-$ ions; 72,142 for the 90 Å box, having 20,756 water molecules and 70 $Na^+$ and 66 $Cl^-$ ions; 163,480 for the 120 Å box, having 53,287 water molecules and 160 $Na^+$ and 156 $Cl^-$ ions; 318,911 for the 150 Å box, having 105,073 water molecules and 309 $Na^+$ and 305 $Cl^-$ ions.

For the electrostatic interactions, particle-mesh Ewald (PME) was used with a grid spacing of 1 Å, a relative tolerance of $10^{-6}$ and a cutoff of 10 Å, together with a 10 Å switching for the Lennard-Jones (LJ) interactions. The LINCS algorithm (*Hess et al., 1997*) was used for constraining bonds involving H-atoms. Each system was first energy minimized for 50,000 steps using steepest descent, heated from 0 to 300 K in increments of 10 K in $NVT$ for 300 ps, followed by 500 ps ($NVT$) and 500 ps of $NpT$ equilibration at $p = 1$ atm with a time step of 2 fs. The center of mass of the protein was restrained to the center of mass of the simulation box. The Velocity Rescaling (*Bussi et al., 2007*) (with $\tau = 0.1$ ps) and Parrinello-Rahman (*Parrinello and Rahman, 1981*) methods were used for temperature and pressure control, respectively. The velocity rescaling method is an extension of the Berendsen thermostat to which a stochastic force chosen such as to generate a correct canonical distribution is added (*Bussi et al., 2007*). The MD simulations for all systems were carried out for at least 1 $\mu$s at constant temperature and pressure ($NpT$) at 300 K and one atm with a time step of 2 fs and the $1\sigma$ temperature fluctuations over the 1 $\mu$s trajectories were 0.1 K for the 150 Å box and 0.5 K for the 90 Å box.

Water self-diffusion coefficients $D$ were calculated for box sizes 75, 90, 120 and 150 Å, in the presence and in the absence of Hb, over the entire 1 $\mu$s trajectory. In the absence of the protein, the simulation boxes contained pure water systems (no ions were included). Including ions at physiological concentrations will typically change the water self-diffusion coefficient by 1% to 2% (*Kim et al., 2012*). First, the mean square displacement (MSD) of all oxygen atoms from a set of initial positions was calculated using mass-weighted averages. Then, the diffusion constant was calculated from the

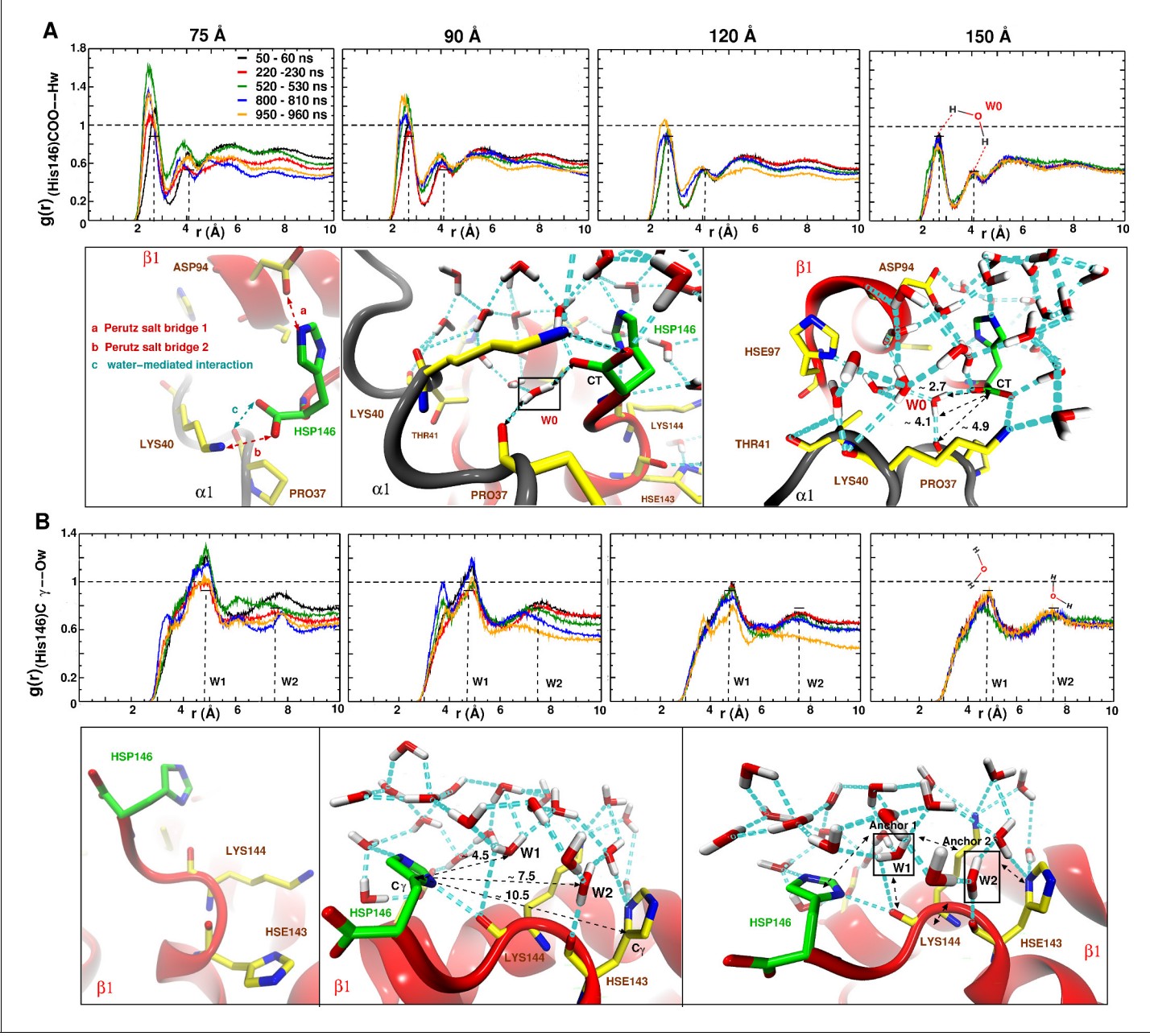

**Figure 6.** Averaged radial distribution functions $g(r)$ between His146 and water for the different box sizes and for simulation windows before, during and after the structural transitions (see *Figure 1*). Analysis for (**A**) the C-terminal (CT) of His146 and Water H (Hw) and (**B**) for C of His146 and water O (Ow). For the largest (150 Å) box no appreciable change in $g(r)$ is found. The two peaks at 2.7 and 4.1 Å in (**A**) and the two peaks at 5 and 7.5 Å in (**B**) are due to a water network as indicated in the sketches. The sketches in A and B represent different views of the same snapshot taken from the MD simulation of the 150 Å box at $t = 960$ ns and is used here as reference to describe the water network because this box represented a stable $g(r)$. Sketches in (**A**) emphasise the water network at the C-terminal of His146, and sketches in (**B**) emphasise the water network at the $C_\gamma$ of His146. Waters W0, W1 and W2 are key structural waters (used as anchor) in the stability of the T state (they are framed with a black line). This network is stable for simulations in the largest box but unstable in the other boxes. His143 is singly protonated ($N\epsilon$) and His146 is doubly protonated ($N\delta$ and $N\epsilon$).
DOI: https://doi.org/10.7554/eLife.35560.019

slope of the mean-squared displacement $D_{\mathrm{PBC}} = \mathrm{Lim}_{t \to \infty} \frac{\partial}{\partial t} \frac{\langle |r(t) - r(0)|^2 \rangle}{6}$ averaged over all water molecules of a particular trajectory. Errors are estimated from the difference of the diffusion coefficients obtained from separate fits over the two halves of the fit interval.

The TIP3P self diffusion coefficient calculated in the simulation by Yeh et al. (using periodic boundary conditions) is $D_{\text{PBC}} = 5.8 \times 10^{-5}$ cm$^2$/s (*Yeh and Hummer, 2004*). However, simulations of water at ambient conditions and a Lennard-Jones (LJ) fluid show that the diffusion coefficients depend on system size (*Allen and Tildesley, 1987*). Thus, the diffusion coefficient corrected for system-size effects is $D_0 = 6.1 \times 10^{-5}$ cm$^2$/s for an infinite system of TIP3P water at 298 K and ambient temperature (*Yeh and Hummer, 2004*). For direct comparison of our values the error $\kappa = L \times (D_0 - D_{\text{PBC}})$ was subtracted from $D_0$. The resulting TIP3P value is $D_{0,\text{corr}} = 5.95 \times 10^{-5}$ cm$^2$/s.

In order to directly compare with the literature (*Yeh and Hummer, 2004*) the 40 Å box was also considered here. The literature value of $D_{\text{PBC}} = 5.65 \pm 0.16$ ($10^{-5}$ cm$^2$/s) compares with $D_{\text{PBC}} = 5.59 \pm 0.013$ ($10^{-5}$ cm$^2$/s) computed here which validates the present simulations.

## Acknowledgements

We thank David Chandler†, to whom we dedicate this paper, for many fruitful discussions. The work in Switzerland was supported by the Swiss National Science Foundation through grants 200021–117810, and the NCCR MUST. The work at Harvard was supported in part by the CHARMM Development Project.

## Additional information

### Funding

| Funder | Grant reference number | Author |
| --- | --- | --- |
| Schweizerischer Nationalfonds zur Förderung der Wissenschaftlichen Forschung | NCCR MUST | Markus Meuwly |
| Schweizerischer Nationalfonds zur Förderung der Wissenschaftlichen Forschung | 200020-169079 | Markus Meuwly |
| Charmm Development Project | | Martin Karplus |

The funders had no role in study design, data collection and interpretation, or the decision to submit the work for publication.

### Author contributions

Krystel El Hage, Conceptualization, Data curation, Formal analysis, Validation, Investigation, Visualization, Methodology, Writing—original draft, Writing—review and editing; Florent Hédin, Prashant K Gupta, Data curation, Formal analysis, Investigation, Writing—original draft; Markus Meuwly, Conceptualization, Resources, Software, Formal analysis, Supervision, Funding acquisition, Validation, Investigation, Visualization, Methodology, Writing—original draft, Project administration, Writing—review and editing; Martin Karplus, Conceptualization, Investigation, Methodology, Writing—original draft, Project administration, Writing—review and editing

### Author ORCIDs

Krystel El Hage (iD) http://orcid.org/0000-0003-4837-3888
Florent Hédin (iD) http://orcid.org/0000-0001-6341-7557
Markus Meuwly (iD) https://orcid.org/0000-0001-7930-8806

### Decision letter and Author response

Decision letter https://doi.org/10.7554/eLife.35560.023
Author response https://doi.org/10.7554/eLife.35560.024

## Additional files

### Supplementary files

• Transparent reporting form
DOI: https://doi.org/10.7554/eLife.35560.020

### Data availability

All data generated or analysed during this study are included in the manuscript and supporting files.

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

# Appendix

DOI: https://doi.org/10.7554/eLife.35560.021

## Structural analyses

The distance between the two $C_\alpha$ atoms of His146 of the two $\beta$ chains is one meaningful coordinate to distinguish the T and the R configuration. The distance between the two $C_\alpha$ atoms of His143 present at the opening of the central cavity can be used to monitor the width of the opening. The RMSD between the $C_\alpha$s relative to the 2DN2-Xray structure was also evaluated.

For the 75 Å box the collapse after 130 ns leads to a concomitant increase in the $C\alpha$-$C\alpha$ RMSD from 2DN2 by more than 1 Å, a repositioning of the $\beta2$ chain relative to the other three chains and a decrease in the degree of hydration of the central channel by up to 20%. Similar observations apply to the 90 (120) Å boxes for which three structural transitions are observed: one after 480 (630) ns (Step 1), the next one after 780 (820) ns (Step 2), and Step 3 after 880 (920) ns. The transitions in the smaller boxes involve different changes (see *Figure 1—figure supplement 2*). Also, the end point structures after 1 $\mu$s are not the same (see *Table 1*).

In order to evaluate and monitor the opening of the central cavity the angle between $\alpha1\beta1$ and $\alpha2\beta2$ sub-units groups is calculated as the angle formed between their planes (*Figure 1—figure supplement 2*). The plane is defined by three atoms: $C\alpha$ of His146$_\beta$, Fe$_\beta$ and Fe$_\alpha$. The smaller the angle the closer are the subunits and the tighter is the central cavity. This affects the number of water molecules that can access the channel. The water count was thus evaluated and confirmed this observation.

The number of water molecules inside the central cylinder of the protein was also analyzed see *Figure 5—figure supplements 3* and *4*. The water count includes only waters in the central cylinder, which was defined as a cylinder of 8.5 Å base area, 22 Å of height and having the four heme moieties as the center of mass, and not waters at the tetramerization interface.

The ion distribution was considered, as well. The concentration of ions was 0.15 mol/L for all box sizes. It is known that the ionic strength of the solution plays a role in the oxygenation behaviour. (*Szabo and Karplus, 1972*) Also, the local ion concentration has an indirect influence by modulating the water activity, which in turn affects allosteric regulation of Hb. (*Salvay et al., 2003*) The ion diffusivity changes from 2.1 to $2.8 \times 10^{-5}$ cm$^2$/s for simulations in the 75 to 150 Å boxes which also reflects the increased diffusivity of the bulk water molecules with increasing box size, see *Figure 4A*. The average number of ions within a 20 Å sphere around the center of mass of Hb is 9.6 (75 Å), 8.5 (90 Å), 8.7 (120 Å), and 9.8 (150 Å). Hence, the ion distribution does not drive the transition directly, which is consistent with additional test simulations carried out, as described below.

## Additional modifications considered

Further modifications for which the stability of the T-state was investigated included (i) strategic placements (by using constraints) of 4 Cl$^-$ ions at the $\beta_1\beta_2$ interface where the ligand 2,3-diphosphoglycerate (DPG) binds, (ii) reduced force constants for the Fe-N$_{His}$ stretching to render the heme group more adaptable to global structural changes of the protein, (iii) placement of the DPG ligand which is known to bind to deoxy-Hb, (iv) modified charges of the hydrogen bonds to stabilize the hinge contacts (*Jones et al., 2014*), (v) different protonation state of His146$_{\beta1}$ and His146$_{\beta2}$ ($\delta-$, $\epsilon-$, and doubly protonated) and (vi) scaling the water-protein nonbonded interactions, specifically the van der Waals well depths. (*Best et al., 2014*) All these simulations were run starting from the T-state in the 75 Å box and were found to become unstable within 70 to 200 ns. The outcome of the last two modifications considered (v and vi) is reported in *Figure 1—figure supplement 6*. The figure shows the $C\alpha$–$C\alpha$ distance between residues His146$_{\beta1}$ and His146$_{\beta2}$ for the T-state Hb in the 75 Å box during 400 ns of 4 MD simulations for the variants (1) protonated His146 (protonated N$_\delta$ and N$_\epsilon$) with (black) and (2) without (red) scaled protein-water interactions; (3) unprotonated His146 (proton on N$_\epsilon$ green) and (4) unprotonated His146 (proton on N$_\delta$ blue) with scaled protein-water

interactions. In all cases the T-state structure becomes unstable to various degrees and decays more or less completely towards the R-state. The red trace in *Figure 1—figure supplement 6* is identical to the black trace in *Figure 1* of the main manuscript.

Particular observations made for these additional simulations are as follows: In the presence 4 $Cl^-$ ions in the cavity where DPG binds the T-state is stable for ~45 ns. Constraining the chloride ions does not significantly extend the time during which the protein is in its T-state. The rationale behind reducing the force constants for the Fe-N bonds is that by softening the internal degrees of freedom the protein as a whole can adapt more easily to perturbations induced by strain. However, even reducing the stretching force constant by a factor of 10 only increases the lifetime of the T-state out to 60 ns. The DPG ligand was placed at the $\beta_1\beta_2$ interface and harmonic constraints were applied to keep it close to this initial position. Despite the present of the ligand, the C$\alpha$–C$\alpha$ distance between His146$_{\beta1}$ and His146$_{\beta2}$ decreases to 20 Å within 20 ns and hence the T-state decays. Recent experimental work (*Jones et al., 2014*) suggested that special hydrogen bond contacts (so called 'hinge' contacts) exist at the $\alpha_1\beta_2$ interface and their presence and absence have been related to the T→R transition. In order to stabilize these contacts (Tyr42–Asp99 and Trp37–Asp94), the partial charges of the hydrogen and acceptor atoms were increased by 10 and 20%, respectively. With this, the T-state lifetime was 50 ns or less, compared to 40 ns if the charges were unmodified. Such tests had also been done for the T-state by constraining the Perutz H-bonds for 100 ns in equilibrium simulations. (*Hub et al., 2010*) However, after removing the constraints, the T-structure decayed again during the following 100 ns simulation. Finally, it had been found that a strengthening of the protein-water van der Waals interaction by 10% is sufficient to recover the correct dimensions of intrinsically disordered and unfolded proteins. (*Best et al., 2014*) Applying this to the apparent instability of the T-state of Hb does, however, not extend its lifetime. After 75 to 120 ns the T-state decays towards the R-state structure (see *Figure 1— figure supplement 6*). We also note that, although in the initial simulation of the 75 Å box with His 146 doubly protonated, the $C_\alpha$-$C_\alpha$ distance between the His 146 residues decreases by only a small amount (from 31 Å to 25 Å in step 1), by other criteria the decaying T-state approaches the R-state (see *Figure 1—figure supplement 2*).

