## [Decision Letter]

Thank you for submitting your article "Valid Molecular Dynamics Simulations of Human Hemoglobin Require a Surprisingly Large Box Size" for consideration by *eLife*. Your article has been reviewed by three peer reviewers, one of whom is a member of our Board of Reviewing Editors, and the evaluation has been overseen by John Kuriyan as the Senior Editor. The reviewers have opted to remain anonymous.

The reviewers have discussed the reviews with one another and the Reviewing Editor has drafted this decision to help you prepare a revised submission.

Summary:

This is a potentially interesting study in which the authors suggest that the stability of the T state of hemoglobin in molecular dynamics (MD) simulations depends critically on the size of the simulation box and that the T state remains stable only with a sufficiently large box containing a relatively large portion of bulk-like solvent. While this observation may bear important practical implications to MD simulation of biomolecules (and polymers in general), its statistical validity needs to be strengthened with further data, and the understanding of its origin requires further analyses.

Essential revisions:

1) The key observation was made based on four simulations in four different simulation boxes. Each simulation has been performed only once. Given the stochastic nature of protein conformational change in MD simulations, repeats of these simulations are essential to show that the observation is of statistical significance. Minimally, each simulation should be repeated once or twice, which should be practically manageable since each simulation is 1 microsecond long. Other simulations should be considered to help establish the key observation. For example, in the cover letter, the authors stated that they "ran simulations starting with a partially decayed (i.e. R-state-like) structure from a 120 ˚A3 box in the 150 ˚A3 box and demonstrated that it progressed toward the T0 structure." These data would be very important, but it is not shown in the manuscript. The authors should show plots of all the distance, angles and RMSDs for this simulation so that it would be easier to assess the significance of their finding. They should also run the same system for an additional 1μs in the 120A box, to provide a fair comparison between the two. Various attempts by the authors to stabilize the T state, which is mentioned in the cover letter but largely not discussed in the main text or the supplemental information, should be reported with some details as well.

2) A possible reason for the instability of the T state would be inaccuracies in the potential energy function, in particular for the Heme group. Also, the 3-point models used to describe water in the simulations are known to provide only a rather rough description of liquid water properties and there is no reason to believe that they would accurately capture the subtle balance of electrostatic and hydrophobic forces that modulate the relative stability of the T and R states. Comparison with previous works is complicated by the use of a different force-field (CHARMM36 (?), instead of the Amber and the Gromos force-fields used in previous works). These issues need to be discussed. Also, the reference for the Charmm36 force-field, which refers to paper for parametrization of the Charmm lipid force-field by mistake, needs to be corrected.

3) The analyses to understand the origin of the possible phenomenon consisted of three parts, respectively concerning hydrogen bonds, diffusion, and radial distribution function of water molecules. The first two analyses showed that, with a larger simulation box, an average water molecule in the simulation increasingly takes up the bulk-water behavior. This is certainly sensible but also to be expected, as in a large box a larger portion of waters are not in the vicinity of the protein. Hemoglobin is a protein that contains a number of cavities and a water-filled channel. It is expected that water molecules in the cavities or in the channel will have a smaller diffusion coefficient and a smaller number of hydrogen bonds with respect to the bulk water. The analysis of the average water diffusion coefficient and number of hydrogen bonds as a function of box size (Figure 3 and 4) is merely showing convergence towards the bulk as the ratio between (water in cavities)/(bulk water) becomes smaller and smaller. The authors need to more clearly state their connection with the T stability of hemoglobin.

4) The authors' underlying hypothesis appears to be that the deviation of water molecules in small boxes from the bulk behavior affects the magnitude of the hydrophobic effect, and that this effect differentially impacts the stability of the two states because the T state is structurally more compact and with less solvent exposed area than the R state. If so, this reasoning should be stated explicitly with necessary data.

5) Figure 5A shows that the water RDF around a spherical solute approaches bulk behavior ~0.5-1 nm away from the solute, suggesting that a periodic box that is ~2 nm larger than the solute should give acceptable result. Figure 5B, however, is apparently at odds with the *g(r*) presented in Figure 5A, as bulk water behavior seems to be achieved only ~2.5 nm away from Hemoglobin. Figure 5—figure supplement 1 presents the whole RDF profile. This RDF calculated in this manuscript appears to be wrong, as there is water in the center of Hemoglobin, so the RDF should have value close to 1 in the center. If this is a mistake, it should be corrected. Did the authors checked that at the beginning of the simulation water was actually placed in the central cavities of the protein and that this water was properly equilibrated?

---

## [Author Response]

Essential revisions:1) The key observation was made based on four simulations in four different simulation boxes. Each simulation has been performed only once. Given the stochastic nature of protein conformational change in MD simulations, repeats of these simulations are essential to show that the observation is of statistical significance. Minimally, each simulation should be repeated once or twice, which should be practically manageable since each simulation is 1 microsecond long. Other simulations should be considered to help establish the key observation. For example, in the cover letter, the authors stated that they "ran simulations starting with a partially decayed (i.e. R-state-like) structure from a 120 ˚A3 box in the 150 ˚A3 box and demonstrated that it progressed toward the T0 structure." These data would be very important, but it is not shown in the manuscript. The authors should show plots of all the distance, angles and RMSDs for this simulation so that it would be easier to assess the significance of their finding. They should also run the same system for an additional 1μs in the 120A box, to provide a fair comparison between the two. Various attempts by the authors to stabilize the T state, which is mentioned in the cover letter but largely not discussed in the main text or the supplemental information, should be reported with some details as well.

As agreed with the Editor we did not repeat these simulations.

2) A possible reason for the instability of the T state would be inaccuracies in the potential energy function, in particular for the Heme group. Also, the 3-point models used to describe water in the simulations are known to provide only a rather rough description of liquid water properties and there is no reason to believe that they would accurately capture the subtle balance of electrostatic and hydrophobic forces that modulate the relative stability of the T and R states. Comparison with previous works is complicated by the use of a different force-field (CHARMM36 (?), instead of the Amber and the Gromos force-fields used in previous works). These issues need to be discussed. Also, the reference for the Charmm36 force-field, which refers to paper for parametrization of the Charmm lipid force-field by mistake, needs to be corrected.

Given that other simulations (Hub et al., 2010 and Yussuf et al., 2012) using different MD codes and force fields find similar instabilities it is unlikely that the FF parametrizations are the prime reason for the thermodynamic instability, see main text, third paragraph. Also, some of the additional tests that were carried out included FF-modifications which, however, did not lead to stabilization in the small water boxes. The reference to the CHARMM36 force field was corrected.

3) The analyses to understand the origin of the possible phenomenon consisted of three parts, respectively concerning hydrogen bonds, diffusion, and radial distribution function of water molecules. The first two analyses showed that, with a larger simulation box, an average water molecule in the simulation increasingly takes up the bulk-water behavior. This is certainly sensible but also to be expected, as in a large box a larger portion of waters are not in the vicinity of the protein. Hemoglobin is a protein that contains a number of cavities and a water-filled channel. It is expected that water molecules in the cavities or in the channel will have a smaller diffusion coefficient and a smaller number of hydrogen bonds with respect to the bulk water. The analysis of the average water diffusion coefficient and number of hydrogen bonds as a function of box size (Figure 3 and 4) is merely showing convergence towards the bulk as the ratio between (water in cavities)/(bulk water) becomes smaller and smaller. The authors need to more clearly state their connection with the T stability of hemoglobin.

In the sixth paragraph of the Results and Discussion it is now clarified that if the water self diffusivity and the hydration network resemble that of bulk water, T_0_ is stable. Figure 3—figure supplement 1 was added to substantiate this. The shape of the radial distribution functions is rather a consequence than the origin of the T_0_ stability.

4) The authors' underlying hypothesis appears to be that the deviation of water molecules in small boxes from the bulk behavior affects the magnitude of the hydrophobic effect, and that this effect differentially impacts the stability of the two states because the T state is structurally more compact and with less solvent exposed area than the R state. If so, this reasoning should be stated explicitly with necessary data.

We have added a statement to the first paragraph of the Results and Discussion which makes clear what is in the papers of Chothia et al. (1976) and Lesk et al. (1985) concerning the role of the hydrophobic effect in stabilizing the T_0_ state. What we do observe is, that the T_0_ state is thermodynamically stable if the water box is sufficiently large such that solvent water behaves as bulk water as judged from the self diffusion coefficient (see Figure 4) and the hydrogen bonded network (see new Figure 3—figure supplement 1).

5) Figure 5A shows that the water RDF around a spherical solute approaches bulk behavior ~0.5-1 nm away from the solute, suggesting that a periodic box that is ~2 nm larger than the solute should give acceptable result. Figure 5B, however, is apparently at odds with the g(r) presented in Figure 5A, as bulk water behavior seems to be achieved only ~2.5 nm away from Hemoglobin. Figure 5—figure supplement 1 presents the whole RDF profile. This RDF calculated in this manuscript appears to be wrong, as there is water in the center of Hemoglobin, so the RDF should have value close to 1 in the center. If this is a mistake, it should be corrected. Did the authors checked that at the beginning of the simulation water was actually placed in the central cavities of the protein and that this water was properly equilibrated?

There are ∼ 130 water molecules in the central cavity of Hb, see new Figure 5—figure supplements 2 and 3. This is from two different analyses. One involves explicit counting and the other one integration of *g(r*). The time series confirm that the central cavity is filled and that the two counts are consistent.